# Cellulose Nanofibrils/Xyloglucan Bio-Based Aerogels with Shape Recovery

**DOI:** 10.3390/gels7010005

**Published:** 2021-01-05

**Authors:** Samuel Mandin, Samuel Moreau, Malika Talantikite, Bruno Novalès, Jean-Eudes Maigret, Bernard Cathala, Céline Moreau

**Affiliations:** 1UR1268 BIA, INRAE, 44300 Nantes, France; smandin9@gmail.com (S.M.); sam.moreau7@hotmail.com (S.M.); malika.talantikite@inrae.fr (M.T.); jean-eudes.maigret@inrae.fr (J.-E.M.); bernard.cathala@inrae.fr (B.C.); 2UR1268 BIA, BIBS Facility, INRAE, 44300 Nantes, France; bruno.novales@inrae.fr

**Keywords:** cellulose nanofibrils, polysaccharides, rheology properties, hydrogels, freeze-casting, porous microstructure, water absorption capacity

## Abstract

Bio-based aerogels containing cellulose nanofibrils (CNFs) are promising materials due to the inherent physical properties of CNF. The high affinity of cellulose to plant hemicelluloses (xyloglucan, xylan, pectin) is also an opportunity to develop biomaterials with new properties. Here, we prepared aerogels from gelled dispersions of CNFs and xyloglucan (XG) at different ratios by using a freeze-casting procedure in unidirectional (UD) and non-directional (ND) manners. As showed by rheology analysis, CNF and CNF/XG dispersions behave as true gels. We investigated the impact of the freezing procedure and the gel’s composition on the microstructure and the water absorption properties. The introduction of XG greatly affects the microstructure of the aerogel from lamellar to cellular morphology. Bio-based aerogels showed high water absorption capacity with shape recovery after compression. The relation between morphology and aerogel compositions is discussed.

## 1. Introduction

Biopolymer-based aerogels, also named bio-aerogels, are particularly attractive as they are fully or partly biodegradable, generally non-toxic, and can be produced from sustainable, renewable sources [1,2,3,4]. Bio-aerogels usually have densities from 0.02 to 0.25 g cm^−3^ and display a wide range of microstructural properties in terms of porosity and morphologies depending on the bio-polymer and the process route. They are thus promising bio-based materials for a wide range of potential applications such as thermal conductivity, super absorption, drug carriage, oil spillage cleanup, biodegradable solid matrices as drug delivery systems, and 3D cellular scaffolds for tissue engineering [4,5,6]. Among all polysaccharides (alginate, chitosan, starch, cellulose and cellulose derivatives, proteins, etc.) exploited for bio-based aerogels, nanocellulosic materials such as cellulose nanocrystals (CNCs) or cellulose nanofibrils (CNFs) derived from plant cell walls have received significant academic and industrial attention due to their potential to replace petroleum-based alternatives [7,8]. 

CNC and CNF exhibit many unique characteristics including relatively high specific surface area, excellent mechanical properties and numerous hydroxyl groups allowing chemical modification such as TEMPO-mediated oxidation, carboxymethylation, or polymer grafting. Furthermore, CNF and CNC both form hydrogels with increasing concentration (ca. 1–10% *w*/*v*, depending on composition) through the formation of percolating networks [1]. In particular, the CNF gel network is formed by physical interactions by nanofibrils entanglements and nanofibril–nanofibril interactions through hydrogen bonds and van der Waals forces. CNF hydrogel can thus form highly flexible, interconnected and porous structure after freeze-drying processes [9].

Nanocellulose-based aerogels are typically prepared by a freeze-casting procedure consisting of a freezing step of the aqueous (both from liquid or gel state) nanoparticle suspensions followed by the sublimation of the solvent [10,11]. The final properties of the aerogel are strongly dependent on the original suspension’s properties (chemical modification, concentration, viscosity) and on the freeze-casting parameters (freezing temperature, temperature gradient, freezing setup) [12,13]. Several strategies were used to enhance cellulose aerogels’ performance including charge surface modification, chemical cross-linking, cation-induced gelling and physical cross-linking.

By controlling the formation and the orientation of ice crystals’ nucleation and growth, different porosities and architectures can be designed, such as lamellar, honeycomb, cellular, etc., in order to optimize the final material properties for applications in many fields such as (super)-absorbents, water filtration, oil remediation and thermal insulation [14,15].

Freeze-casted CNF and CNC aerogels usually exhibit a lamellar pore structure and a highly oriented structure can be obtained using directional freezing processes [16,17]. Several works have investigated the impact of freeze-casting conditions on morphological and mechanical properties of CNC- and CNF-based aerogels [18,19,20,21,22]. In their study, Darpentigny et al. [23] compared freeze-casted aerogels from different nanocellulose sources (tunicate and cotton CNC, CNF) and showed that aerogels from tunicate CNC exhibit better mechanical performance due to the honeycomb structure and structural properties of these CNCs (high crystallinity and aspect ratio). 

Another interesting property of nanocellulose aerogel is the ability to absorb water. Yang et al. [24] prepared ultralightweight and highly porous aerogels from chemically cross-linked CNC suspensions and showed that such CNC aerogels can absorb significant amounts of water. Chemically cross-linked aerogels also display a great shape recovery ability in water after compression [24]. However, to improve the water absorbance capacity of nanocellulose aerogels, a solvent exchange procedure and/or chemically cross-linked nanocellulose are usually used to compensate for the inherent fragility of CNF and CNC aerogels, which have pore size in the µm range [24,25,26,27]. 

Among all approaches to enhance performance of nanocellulose aerogels, mixing nanocellulosic materials with other polysaccharides of the plant cell wall, such as hemicelluloses, xyloglucans [16,18,28], xylans [29,30], galactoglucomannan [31], etc., have been exploited to form biomass-based aerogels. Hemicelluloses are well known to have high affinity to cellulose and exhibit a range of interactions with cellulose due to their large structural and chemical variations [32,33,34]. Several works deal with cellulose–hemicellulose interactions in model systems such as films, hydrogels, composites, and aerogels with novel functionalities could be added to the original cellulosic materials [33,35].

Among hemicelluloses, xyloglucans are known to bind particularly efficiently to cellulose surfaces through van der Waals and hydrogen bonding interactions. For several years now, our research group has focused on cellulose-xyloglucan interactions, and demonstrated that XG absorbs irreversibly on cellulose surface, and can adopt different conformations as a function of the XG molar mass and XG/cellulose ratio [36,37]. Indeed, at low cellulose/XG ratio, XG covers all the cellulose surface available and thus XG acts a cross-linking agent, while at higher ratio the cellulose surface is oversaturated, increasing the polymer crowding limiting interaction between cellulose surfaces. By taking advantage of cellulose-XG interactions, we developed biocomposites with new functionalities such layer-by-layer (LbL) films for enzyme detection [38,39], CNC-XG hydrogel with thermos-responsive properties [40], and more recently CNC-XG aerogels exhibiting promising mechanical and water-resistant properties [16]. Incorporation of XG into nanocellulose suspensions before aerogel preparation leads to morphological changes and enhances their mechanical properties as function of the cellulose/XG ratio as compared to pure nanocellulose aerogels [16,18,28]. 

Inspired by our precedent works on CNC/XG systems [16,36], we explored the microstructural organization and water absorbent capacity properties of CNF and CNF/XG aerogels. Rheological properties of initial aqueous CNF/XG dispersions by varying CNF/XG mass ratio from 100:0, to 95:5 and 60:40 (wt %:wt %) were first evaluated to determine the rheological properties of CNF/XG mixtures. Freezing conditions, i.e., anisotropic (unidirectional (UD)) and isotropic (non-directional, ND), of gel mixtures were conducted to investigate the impact of nanoparticles’ orientation on the microstructure and water absorption capacity of aerogels. We demonstrated the shape recovery capacity of compressed aerogels in water and discussed their water absorption capacity as a function of the composition and microstructure of the CNF and CNF/XG aerogels. 

## 2. Results and Discussion

### 2.1. Rheological Properties of CNF and CNF/XG Mixtures

The rheological behavior of CNF dispersion at 2 wt % and 3 wt % and the corresponding CNF/XG mixtures at ratios of 100:0, 95:5 and 60:40 (corresponding to XG/CNF concentration ratios of 0, 0.05, and 0.67 mg XG mg^−1^ CNF) by keeping constant the final content at 2 wt % and 3 wt %, were investigated with a focus on the evolution of the storage (G′) and loss (G′’) moduli (Figure 1A,B) and complex viscosity (Figure 1C) as a function of frequency (s^−1^). 

As displayed in Figure 1A, in case of CNF100 at 2 wt %, storage modulus (G′) was higher than loss (G″) modulus in the majority of used frequency range, describing a solid-like behavior; indeed at very high frequency we observed a crossover point after which G″ overcame G′. This behavior was already described in literature: Chen et al. [41] reported the presence of two crossovers in case of low concentration CNF suspension (0.1 wt %), which normally denoted a reorganization or change in structure/phase. They have clearly explained that the phenomenon reveals a partial gelation is the first event for the G′ and G″ crossover and the second is a sort of ‘‘disentanglement’’ or gel destruction as frequency increases.

In case of CNF100 at 3 wt % (Figure 1B), we observed that G′ was higher than G″ in all the range of frequencies measured (solid-like behavior). The network formed at this concentration was high enough to resist higher frequencies. Increasing CNF from 2 wt % to 3 wt % led to increase in moduli values for almost one decade order, probably due to the increase of the number of physical entanglements [41]. This increase in physical entanglement was evidenced by the increase in zero-shear viscosity values (Figure 1C): 69 Pa.s in case of 2 wt % while it was 1457 Pa.s for 3 wt % CNF.

For both 2 wt % and 3 wt % total content, the XG fraction increasing in the complex led to the increase in hydrogel strength as reflected by G′ and G″ increasing. XG adsorption on CNF led to the increase of volume fraction in the suspension [22], thus to the entanglements of CNF/XG complexes leading to the increase of complex viscosity (Figure 1C). In more detail, for 2 wt % suspension, viscosity values increased from 69,928 to 4423 Pa.s for CNF100, CNF95:XG5 and CNF60:40, respectively. The same trend was observed for 3 wt % suspensions with 1457, 2596 and 5903 Pa.s for CNF100, CNF95/XG5 and CNF60:40, respectively.

For all samples, the complex viscosities linearly decreased with frequency denoting that all samples exhibited a shear-thinning behavior.

We found that CNF at 2 wt % possessed a viscous dominant behavior while the 3 wt % CNF behaved as a true gel, as previously observed by Mendoza et al. (2018), who obtained true gels with TEMPO-CNF at 0.29 wt %. Rheological properties of CNF are primarily affected by CNF concentration. TEMPO-CNF used in this study possessed high charge density (carboxyl groups 1.4 mmol/g CNF) so that repulsive forces among fibrils was high and concentrated CNF dispersion was needed for gelling effect.

When XG was added to CNF dispersion at 95:5 and 60:40, all CNF/XG dispersions displayed a gel-like behavior indicating that, as CNF content decreased, XG reinforced the hydrogel network. This gelation phenomenon was due to the coverage of CNF surfaces by XG and the physical cross-linking between CNFs as already observed for CNC-XG systems [16,18]. 

### 2.2. Characterization of CNF and CNF/XG Aerogels

Lightweight and highly porous CNF and CNF/XG aerogels were prepared using unidirectional (UD) and non-directional (ND) freeze-casting methods by freezing mixtures at −20 °C followed by freeze-drying (the freezing procedures are schematically displayed in the Appendix A). In the UD process, freezing gradient occurred from the bottom to the top of the gel, during which ice crystals nucleated and grew vertically leading to nanoparticles’ rejection from the solidification front. From the ND freezing process, ice crystals nucleated and grew in isotropic direction, from outer surface to the core of the gel network. Finally, ice crystal sublimation using freeze-drying resulted in porous materials with pore shape corresponding to ice crystal structure formed during freezing [42]. Aerogels of 23.6 ± 1.3 kg m^−3^ and 32.5 ± 1.6 kg m^−3^ with an average apparent porosity of 98.5% (±0.1) and 97.8% (±0.1), respectively, for all compositions were successfully obtained (Table 1). 

In order to investigate the impact of the freezing procedure and of the dispersion composition, CNF and CNF/XG aerogels prepared from unidirectional (UD) freezing to form anisotropic porous structure were compared to isotropic aerogels (ND freezing) of the same composition. 

#### Morphology of Aerogels 

Figure 2 shows the SEM images of CNF and CNF/XG aerogels at 23.6 and 32.5 kg m^3^ from anisotropic (unidirectional, UD) and isotropic (non-unidirectional, ND) freezing with 100/0, 95/5 and 60/40 CNF/XG compositions. SEM observations revealed different microstructures of the aerogels as a function of the freezing procedure and when XG was present. It is well-known that aerogel microstructure depends on several factors such as the freezing process, temperature gradient, and suspension viscosity [12,20]. In this present study, as freezing temperature was kept at −20 °C during 1.5–2 h for all preparations, the morphology changes were mainly due to the freezing condition (UD or ND) and to the dispersion viscosity and network. 

A lamellar structure was observed in the cross-section for all CNF aerogels from UD and ND freezing processes (Figure 2a’,e’,h’,k’). This lamellar structure was already observed for CNF- and CNC-based aerogels [16,17,30,43,44,45]. UD CNF aerogels displayed well-oriented lamellar structure in the freezing direction (Figure 2a,e) (as indicated by the red arrow in Figure 2e) with pores between CNF walls observed in the cross-section (Figure 2a’,e’). Orientation of lamellar structure was due to the growing of the ice crystals in the vertical direction throughout the UD freezing process that rejected and concentrated the CNF particles between ice columns [42]. After freeze-drying, lamellar structures of self-assembled cellulose nanofibrils were formed where vertically aligned pore walls were a replica of the ice crystals. 

For ND CNF aerogel (Figure 2h,h’,k,k’), in longitudinal section and cross-section, at least two distinct orientations of nanofibril lamellar structures could be observed, confirming that ice crystals nucleate and grow in isotropic direction, from outer surface to the core of the hydrogel network, as previously observed [46]. Interconnected CNF lamellae were clearly visible from ND samples (Figure 2h’), suggesting that not all nanofibrils were concentrated into walls during the isotropic growth of ice crystals. Smooth CNF walls with thickness of about 1–4 µm were observed for both UD and ND aerogels at 23.6 and 32.5 kg m^−3^, with mainly 4 µm wall thickness for ND aerogels, indicating a great aggregation of nanofibrils during the freezing process and CNF surface’s covering by XG. 

When XG was added to the initial CNF dispersion, the resulting aerogel pore morphology clearly changed (Figure 2). In this case, a cellular morphology for both UD and ND freezing was observed as already described for CNF/XG and CNC/XG systems by Sehaqui et al. [28] and Jaafar et al. [16], respectively. This difference in morphology could be attributed to CNF-XG assembly formed before the freezing step, hindering the ice crystal growth in the vertically direction. 

UD CNF/XG aerogels displayed a macroporous network with pores oriented in the freezing direction and different pore sizes as a function of the suspension viscosity. A pore size of 30–90 µm was measured for 23.6 kg m^−3^ aerogels (Figure 2b,c’) while denser aerogels presented a larger pore size distribution with macropores (50–140 µm) (Figure 2f,g’). This would be imparted by the higher viscosity of hydrogels at 3 wt % limiting the ice crystal growth, as previously observed in our CNC/XG systems [16] and in other systems. Kam et al. [18] found aerogels with wider pores due to the increase of the viscosity when XG was added to CNC dispersion. By changing the molecular structure of xylan using enzyme treatment, thus reducing the viscosity of the xylan solution, Köhnke et al. [30] also observed a pore morphology change of xylan foam. While more homogeneous pore size was mainly observed for CNF/XG aerogels at lower density, a larger pore size distribution was observed for denser aerogels prepared from more viscous suspensions (Table 1). Because of the increase of gel density and viscosity, the volume of the solid content increased, subsequently affecting the pore size and pore size distribution [14,47].

ND CNF/XG aerogels displayed open and closed pores (Figure 2i,i’–m,m’) that were not visible from UD aerogels. During ND freezing, there was no preferred crystal growth, which resulted in random orientation of the pores. For ND aerogels at 32 kg m^−3^, open pores with 20–120 µm in size were similar in size to those observed with UD aerogels (Table 1). Larger sizes were measured for closed pores with a higher size dispersion: 160–280 µm for CNF95/XG5 and 190–320 µm for CNF60XG40. This pore morphology was due to the isotropic growth combined with the strong and viscous CNF-XG network. 

These observations demonstrate that the formation of a viscous 3D CNF-XG network has an impact on freezing front velocity and, consequently, the entrapment and concentration of cellulose nanofibrils between ice crystal “columns” during the freezing procedure could be limited and hindered. Furthermore, the presence of XG clearly affects the final structure by acting as cementing agent between cellulose surfaces and/or by tethering CNF fibers through loops and tails [36]. 

### 2.3. Water Absorption Capacity and Shape Recovery

After compression at 95% strain, aerogel structure did not fracture but a denser structure was obtained at about 70–80% of the original thickness (Figure 3). 

When water is added to compressed aerogels, CNF and CNF-XG aerogels immediately absorbed water (1–2 mL in 20 s) and recovered their initial shape. Aerogels absorbed water at around 30–40 times of their own weight corresponding to relative absorbed volume of 88–100% water (Table 1). From values in Table 1, higher water absorption capacity was observed for 2 wt % UD aerogels (~40 g H_2_O g^−1^ aerogel) than for denser aerogels (~30 g H_2_O g^−1^ aerogel). Additionally, UD CNF60XG40 aerogels appeared to have slightly lower water capacity than CNF or CNF95/XG5 aerogels. The same observation could be made for ND aerogels, with higher water absorption capacity observed for 2 wt % aerogels. Thus, water absorption capacity decreased as the aerogel density decreased as reflected by porosity values (98.5% for 23 kg/m^3^ vs. 97.8% for 32 kg/m^3^) and it was correlated with the CNF content. 

These absorption capacity values are in the same range of other reported data for nanocellulose aerogels [24,25,27,31]. Cervin et al. [25] prepared cross-linked CNFs aerogels at 20 kg m^−3^ and showed that such aerogel can absorb 38 times their own weight after compression. Superabsorbent nanocellulose-based aerogels were also demonstrated by Yang and Cranston [24], who found high water absorption capacity (160 g g^−1^) for aerogels prepared from aldehyde-functionalized CNC and hydrazide cross-linking. Jiang et al. [26] found a water absorption capacity between 53 g g^−1^ and 92 g g^−1^ for CNF aerogels at 8 kg/m^3^ prepared after freeze-thawing (FT) cycles, and water exchange to *tert*-butanol step. They found that water absorption capacity was inversely correlated with the number of FT cycles. Indeed, during freeze-thawing procedure, water was expelled from the aerogel so that interactions between nanofibrils were reinforced leading to a decrease of apparent porosity with increasing FT cycles. GGM/CNF aerogels exhibit sponge-like behavior with a water uptake capacity of about 30–37 g water g^−1^ aerogel and the water capacity uptake increases as the CNF content increases [31]. These results indicate that the collapsing of pore walls during compression is reversible as aerogels return back to their original form. Zhang et al. [27] explained that the superabsorbent property of CNF/MFC aerogels depends on to the hydrophilic nature of cellulose but also on the force of the crosslinking points. Water penetration in the dry aerogel will create hydrodynamic force, forcing the deformed aerogel structure to expand.

Interestingly, when an aqueous solution of HCl 100 mM was deposited on aerogels, aerogels limited swelling and retained their compressed shape (height: 3.8–4.2 mm for CNF100 and 3.5 mm for CNF60XG40). At acidic pH, carboxylate groups present on the TEMPO-CNF surface became protonated to a larger extent resulting in the reduction of electrostatic repulsion. The protonation of carboxyl groups promoted the formation of hydrogen bonds between nanofibrils so that cohesion between nanofibrils inside aerogels walls was reinforced through a physical cross-linking. This led to less available sites for water molecules’ interaction, preventing penetration of water molecules inside walls and thus the swelling of aerogels. When aqueous NaOH solution (100 mM) was poured on aerogels and observed after about 4 h, a re-swelling process was observed for aerogels but the swelling was slow (4 h) and not complete. 

#### Compression Tests of Wet Aerogels

The swollen aerogels were assessed by compression testing to ascertain their mechanical response. Compression was performed up to 50% of the aerogel height at each cycle and Figure 4 shows stress-strain curves from compression cycles of aerogels at 2 wt %.

As displayed in Figure 4A, wet CNF aerogel kept its integrity after 15 compression cycles. During a compression cycle, water was expelled from the bottom of the aerogel structure and the CNF aerogel structure became denser as not all water content was reabsorbed during the unloading step. Up to 6 compression cycles, a slowly increasing stress response was observed. Between the 8th and 12th compression cycles, an exponentially increasing stress was observed above 70% strain. In these cases, higher compression strength was needed to achieve the densification regime; at 85% strain a compressive stress of 5.84 kPa was needed at the 10th compression cycle. This value is in the same order to that found by Yang and Cranston [24], who found 3.7 kPa at 80% strain for cross-linked CNC aerogel at 2 wt %, and Jiang et al. [26] reported 1.2 kPa for CNF aerogels compressed at 80%. 

At 95/5 ratio (Figure 4B), CNF/XG aerogel exhibited the same behavior up to 10 compression cycles but the exponentially increasing stress was observed above 55% strain and was more progressive as compared to CNF aerogel. The compression stress was 4.44 kPa at 70% strain at the 10th cycle. In contrast, the structure of CNF/XG aerogel at 60/40 ratio was totally disintegrated after 5 compression cycles only (Figure 4C). 

The compression of wet aerogels led to water expulsion from the network and wet CNF aerogel was more mechanically resistant as compared to CNF/XG aerogels. The XG content was negatively correlated to compressive strength of the wet aerogels, as CNF60XG40 was disintegrated after 5 compression cycles while the CNF95XG5 aerogel could bear 10 compression cycles. 

Yang and Cranston [24] explained that, with the expulsion of water during compression, the porous structure starts to collapse, leading to high bearing capacity. Since XG is water soluble and CNF is water dispersible, CNF/XG may lose their integrity in the presence of water at XG excess (60:40, wt %:wt %). Because CNF and XG are polar materials, water acts as plasticizer and weak hydrogen bonding agent by preferentially forming hydrogen bonds with XG, softening the 3D CNF-XG network [24,27,31]. As observed by SEM analysis, UD CNF aerogels displayed lamellar structure while a cellular microstructure was observed for UD CNF/XG aerogels (Figure 2). This lamellar structure preserved the network integrity during compression under a wet state as water-filled pores were oriented in the compression direction. In contrast, the cellular structure of wetted CNF/XG aerogel could be weakened after compression as XG-CNF crosslinks were broken with a more brittle network as the XG content increased. Indeed, at high XG content, a slippery effect of XG could be seen as previously described with CNC-XG systems [16]. 

## 3. Conclusions

This work has shown the impact of gel composition and freezing conditions on the corresponding CNF/XG aerogel microstructure. A change from lamellar to cellular morphology occurred in the presence of XG due to the cross-linking of XG to CNFs. After compression, these bio-based aerogels displayed high water absorption and shape recovery properties, which were mainly governed by electrostatic interactions through surface charge of nanocellulose. The macrostructure of wet bio-aerogels without any chemical pretreatment appeared resistant under compression but XG content and incorporation of CNF dispersion are still under investigation to enhance the compression performance of aerogels. By controlling the porous structure of the 3D network and the charge density on CNF surfaces, such bio-based aerogels could be designed as promising sensitive materials. 

## 4. Materials and Methods 

### 4.1. Materials

2,2,6,6-tetramethylpiperidinyl-1-oxyl (TEMPO)-oxidized cellulose nanofibrils (CNF) (freeze-dried solid, sodium form, 1.4 mmol—COONa/g dry CNF) were purchased from the University of Maine (Orono, ME, USA); they were obtained by mechanical disintegration through applying a chemical pre-treatment, namely surface carboxylation of cellulose nanofibrils using TEMPO. Tamarind seed xyloglucan (**XG**) was purchased from DSP Goyko Food & Chemical (Osaka, Japan). Macromolecular characterization of XG was previously determined as Mw = 10.28 × 10^5^ g mol^−1^ [37].

### 4.2. Preparation of CNF and XG, CNF/XG Hydrogels

Aqueous CNF dispersions were prepared at 20 and 30 g/L by dissolving CNF under vigorous stirring over 24 h at room temperature. XG solutions were prepared at 20 and 30 g/L by dissolving XG powder in water at 50 °C and stirring for 24 h at room temperature. Both CNF and XG preparations were kept at 4 °C before use. XG solution was added to CNF suspension to obtain final CNF/XG mass ratios; i.e., 95:5 and 60:40 (wt %/wt %) to a final and constant density of 2 wt % and 3 wt % (Table 1). The mixture was stirred under vortex before pouring into syringe for the freezing step (cf Section 4.4). Samples were noted as CNF100, CNF95/XG5 and CNF60/XG40. 

### 4.3. Rheological Characterization of Hydrogels

All rheological measurements were performed using a stress-controlled rheometer AR-2000 (TA Instruments) equipped with truncated cone (40 mm diameter, 2° cone) at 25 °C. 

The elastic (*G′*) and viscous (*G″*) moduli were measured within the linear response regime, meaning that the values of *G′*, *G″* were not dependent on the applied stress; this allowed the sample to be probed in its equilibrium state, without any shear effect on the structure. From the values of *G′* and *G″* the value of complex viscosity $\left|\eta^{*}\right|$ was determined as
(1)$$\left|\eta^{*}\right|=\;\frac{f}{2\pi }\;\sqrt{{G^{\prime }}^{2}+{G^{\prime \prime }}^{2}}$$
where *f* was the frequency in Hz. Samples were covered with paraffin oil to prevent drying.

### 4.4. Aerogel Synthesis 

Aerogels were prepared by using non-directional (ND) or unidirectional (UD) freeze-casting methods (Appendix A); 3–5 mL hydrogels were poured into a polypropylene syringe mounted of the top surface of a metallic aluminum bar for UD freezing or sealed with plug for ND freezing. Hydrogel was kept at 4 °C overnight before freezing step. Different freezing conditions were used: (i) for the UD freezing method, hydrogel was frozen by placing the metallic bar vertically into ice/NaCl bath at −20 °C to allow vertical ice crystal growth; (ii) for the ND freezing method, samples were placed in a freezer at −23 °C to allow ice crystal growth in an isotropic manner. After the freezing step over 1.5–2 h, samples were subjected to freeze-drying (SRK GT2, Systemtechnik GMBH, Germany) to allow the frozen water in the samples to sublimate directly from the solid phase to the gas phase. The freeze-drying process was carried out at −90 °C and 3 × 10^−2^ mbar and left to proceed for 48 h, allowing for the recovery of a dried lightweight solid. At least 6 aerogels of similar composition were synthetized. The resulting aerogels, in the cylindric shape, were cut to 10 × 10 × 10 cm^3^ cubic geometry using a razor blade. Apparent densities (ρ_aerogel_) of the aerogels were calculated by weighing the samples and measuring dimensions using calipers. Apparent porosity (%) was estimated from the ρ_aerogel_ and considering ρ_cellulose_ and ρ_XG_ to be 1500 kg m^−3^ and 1400 kg m^−3^, respectively. 

### 4.5. Scanning Electron Microscopy (SEM) Analysis

Images of the aerogels were obtained using a field emission gun scanning electron microscope (Quattro S, Thermo Fischer Scientific). The specimens were fixed onto a metal stub with conductive carbon tape and sputter-coated with 5 nm platinum layer by an ion-sputter coater (LEICA EM ACE600, Germany) and observed in HI-VAC mode under 5 kV acceleration and 30 µA probe current. 

The estimated pore sizes and wall thicknesses were measured using ImageJ^®^ software.

### 4.6. Water Absorption Capacity after Compression

Cubic CNF and CNF/XG specimens of 1 cm^3^ were compressed using a MTS SYNERGIE 100 machine using a load cell of 100 N before measuring water absorption capacities. First, aerogels were compressed in dry state to a 90% compressive strain at 3 mm per minute to the initial sample height (i.e., to the freeze-casting direction for UD samples). Then, about 1.5 mL of deionized water or 100 mM HCl was deposited on the compressed aerogel and after 20 s, excess of water was removed by filter paper and the aerogel was weighed again. 

After water swelling, aerogels were compressed up to 50% of their height at 2 mm per minute. The compressed aerogels were then submitted to several analysis cycles which consisted of compression testing up to 50% of their new hydrated height followed by 3–4 min unloading step.

Compressed and swelled aerogel specimens were weighed and placed onto the plate alongside a polylactide (PLA) cube as template of the initial cubic aerogel (1 cm^3^). Dimensions of compressed and swelled aerogels were measured from pictures using ImageJ^®^. 

Water absorption capacity (*Q*) of aerogels was calculated from:(2)$$Q\left(g/g\right)=\left(\frac{m_{abs}-m_{CP,\;dry}}{\rho_{liq}}\right)/m_{dry}$$

With *m_CP,dry_* the mass of the compressed aerogel in dry state, *m_abs_* the mass of the aerogel after water absorption and *ρ**_liq_* the bulk density of water (0.9882 g/cm^3^ at 20 °C). 

The shape recovery percentage was calculated as:(3)$$Shape\;recovery\;percentage\;\left(\%\right)=h_{final}/h_{initial}\times 100\%$$

With *h_final_* and *h_initial_* corresponding to the original height and the final volume of the aerogel, respectively. 

## Figures and Tables

**Figure 1 gels-07-00005-f001:**
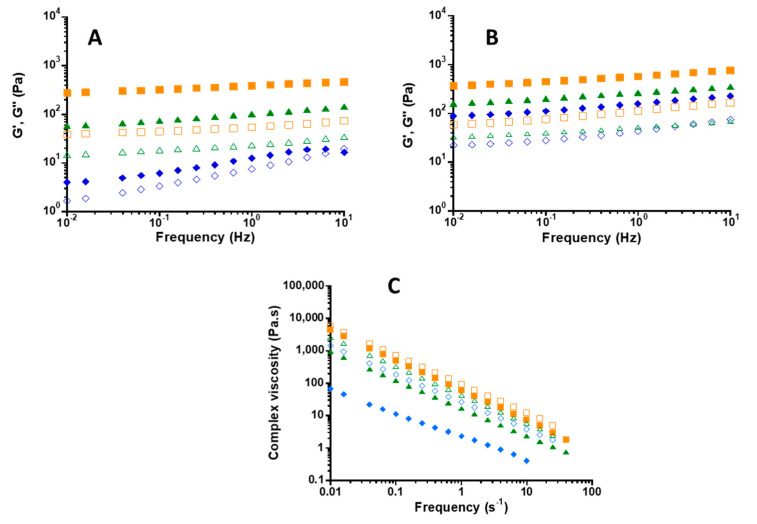
(**A**,**B**) G′ (full symbols) or G″ (empty symbols) (Pa) vs. frequency (Hz) for CNF/XG ratio: 100:0 (blue), 95:5 (green), 60:40 (orange) with final concentrations at 2 wt % (**A**) and at 3 wt % (**B**). (**C**) Complex viscosity (Pa.s) vs. frequency (s^−1^) for CNF/XG ratio: 100/0 (blue), 95/5 (green), 60/40 (orange) with final concentrations at 2 wt % (full symbols) and 3 wt % (empty symbols), respectively.

**Figure 2 gels-07-00005-f002:**
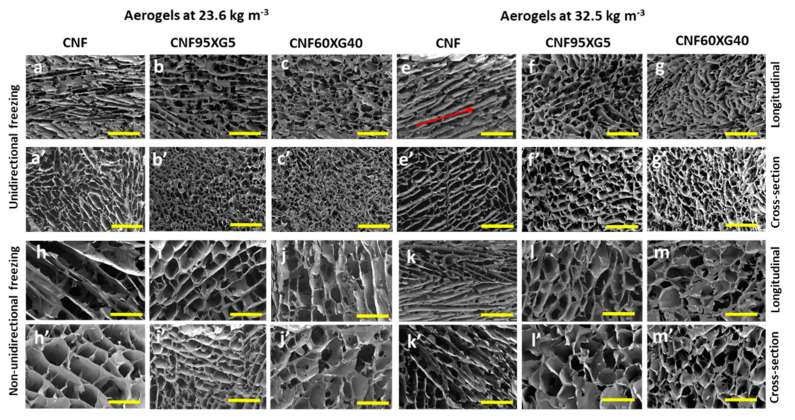
SEM images of the aerogels at 23.6 and 32.5 kg m^−3^ with the three compositions CNF100, CNF95XG5, CNF60XG40 prepared by unidirectional (**a**–**g**; **a’**–**g’**) and non-directional (**h**–**m**; **h’**–**m’**) freezing processes. SEM images from longitudinal (**a**–**g**; **h**–**m**) and cross-section (**a’**–**g’**; **h’**–**m’**) views. Yellow scale bars are of 500 µm.

**Figure 3 gels-07-00005-f003:**
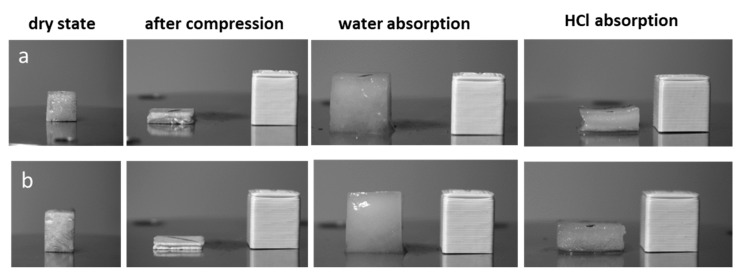
CNF aerogel (2%, UD freezing) at (**a**) 100:0 CNF/XG and (**b**) 60:40 CNF/XG in dry cubic shape, after compression, and after water absorption or after HCl 100 mM absorption. Cubic specimen in polylactide (PLA) at right is used as reference.

**Figure 4 gels-07-00005-f004:**
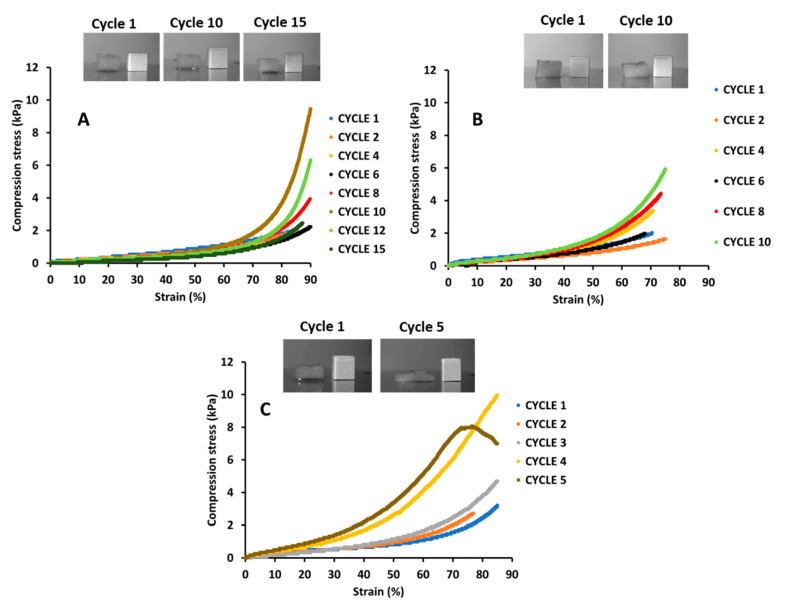
Compressive stress (kPa) vs. strain (%) of wet 2 wt % aerogels (**A**) CNF100 and (**B**) CNF95XG5 and (**C**) CNF60XG40.

**Table 1 gels-07-00005-t001:** Characteristics of CNF and CNF/XG aerogels prepared from the aqueous dispersions.

Density of the Aqueous Dispersions	Aerogels	Apparent Density (kg/m^3^)	Apparent Porosity (%)	Pore Size (µm)	Abs. Capacity (g H_2_O g^−1^ Aerog.)
	UD aerogels				
2 wt %	CNF	23.4 ± 2.1	98.4 ± 0.1	-	42.1 ± 4.4
	CNF95XG5	22.7 ± 2.0	98.6 ± 0.4	40–90 µm	42.9 ± 3.4
	CNF60XG40	24.1 ± 0.9	98.4 ± 0.1	30–70 µm	36.9 ± 2.0
	CNF	32.2 ± 1.9	97.9 ± 0.0	-	31.6 ± 1.4
3 wt %	CNF95XG5	30.2 ± 2.8	98.0 ± 0.3	50–140 µm	32.7 ± 3.2
	CNF60XG40	30.6 ± 1.9	97.8 ± 0.1	60–120 µm	29.2 ± 3.2
	ND aerogels				
	CNF	24.7 ± 0.7	98.4 ± 0.1	-	37.3 ± 1.4
2 wt %	CNF95XG5	24.6 ± 1.1	98.6 ± 0.2	30–90 µm	38.9 ± 1.4
	CNF60XG40	23.5 ± 1.4	98.4 ± 0.1	50–110 µm	39.8 ± 1.4
	CNF	31.7 ± 0.9	97.9 ± 0.1	-	29.2 ± 1.6
3 wt %	CNF95XG5	31.1 ± 4.5	97.9 ± 0.3	20–120 µm	26.8 ± 0.7
	CNF60XG40	32.6 ± 1.9	97.8 ± 0.1	20–120 µm	27.6 ± 0.9

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
