# Peer review of "Cellulose Nanofibrils/Xyloglucan Bio-Based Aerogels with Shape Recovery"

_gels, 2021, doi:10.3390/gels7010005_

Round 1

Reviewer 1 Report

The article is perfectly fitting the journal scope and is written in good English. The experiments were well performed and the results explained. There are however, some errors that occurred during writing, which should be corrected before publishing. Some suggestions are bellow:

Line 109: The units for “Complex viscosity” in the figure 1C should be (Pa.s), please correct.

Line 129: Replace “Figure 3C” by “ Figure 1C”.

Table 1: The authors should specify the differences between aerogels, such as 2 %wt and 3 %wt NFC, the table is not clear.

CNF and NFC is used over the text, the authors should decide which to use and be consistent.

Why did the authors studied the mixture ratios at 100:0, 95:5 and then 60:40? Why not an intermediate value, like 80:20, which would be necessary to draw conclusions about the influence of ratio.

Line 239: “CNF100/XG5 aerogels” should be “CNF95/XG5 aerogels”, please correct

Line 273: “Figure 6” should be “Figure 4”, please correct

Line 289: Remove one bracket after 2 %wt

Line 289: Reference after “Jiang et al” should be placed

Line 396: Authors should place “,” after “Mandin” and remove after “Samuel”

Line 399: Remove “Celine” after “Moreau”

Line 401: Remove one “and” word

Author Response

Reviewer 1

Comments and Suggestions for Authors

The article is perfectly fitting the journal scope and is written in good English. The experiments were well performed and the results explained. There are however, some errors that occurred during writing, which should be corrected before publishing. Some suggestions are bellow:

Line 109: The units for “Complex viscosity” in the figure 1C should be (Pa.s), please correct.

The correction of the units was done in the Figure C.

Line 129: Replace “Figure 3C” by “ Figure 1C”.

Figure 3C was replaced by Figure 1C (Line 131 now)

Table 1: The authors should specify the differences between aerogels, such as 2 %wt and 3 %wt NFC, the table is not clear.

CNF/XG Aerogels were prepared from the corresponding aqueous dispersions at a final density of 2 wt% and 3 wt%. The apparent density of each aerogel was then calculated by weighing the samples and measuring dimensions. A column was added in Table 1 in order to specify the final density (wt%) of the aqueous dispersions and avoid ambiguity with the apparent density of the aerogels.

CNF and NFC is used over the text, the authors should decide which to use and be consistent.

The abbreviation “CNF” for Cellulose NanoFibrils was choosen and put all over the text.

Why did the authors studied the mixture ratios at 100:0, 95:5 and then 60:40? Why not an intermediate value, like 80:20, which would be necessary to draw conclusions about the influence of ratio

The choice of the two CNF/XG ratios is based on our previous study (Dammak et al, Biomacromolecules, 2015, 16, 589-596) that investigate the XG adsorption isotherm onto CNC surface. At low content of XG (95/5), the CNF surface are not covered by XG while at higher XG content, CNF surface are supposed to be saturated by XG since saturation limit should be expected to range between 80/20, 70/30 ratios. Consequently, we expected to have contrasted behaviors by comparing CNF/XG mixtures at 95/5 and 60/40. This explanation was added in the introduction (Line 82).

Line 239: “CNF100/XG5 aerogels” should be “CNF95/XG5 aerogels”, please correct

The correction is done in the text (Line 241 now)

Line 273: “Figure 6” should be “Figure 4”, please correct

The correction is done in the text (Line 275 now)

Line 289: Remove one bracket after 2 %wt

Brackets are removed.

Line 289: Reference after “Jiang et al” should be placed

The reference was inserted correctly in the text (Line 291 now)

Line 396: Authors should place “,” after “Mandin” and remove after “Samuel”

Line 399: Remove “Celine” after “Moreau”

Line 401: Remove one “and” word

As recommended by the rewiever 3, the section « Author Contributions » has been rewritten by using the recommended statements of the Gels Journal.

Reviewer 2 Report

  • SEM image magnification bars are not clear. Author should correct it in all images.
  • Author should provide BET data for prepared aerogel samples.
  • English of the manuscript need to be revised carefully.

Author Response

Reviewer 2

Comments and Suggestions for Authors

  • SEM image magnification bars are not clear. Author should correct it in all images.
  • The scaling bars in SEM images of the figure 2 correspond to 50 µm for all images and have been all increased in thickness and highlighted with a yellow bar for a better visualization.
  •  
  • Author should provide BET data for prepared aerogel samples.

BET analysis rely on gas (typically nitrogen) adsorption-desorption isotherms and allow the determination of the specific surface area and, as associated information, size of mesopores (from few nanometers to tens of nanometers). This analysis is relevant when aerogels display high specific surfaces. In our experiments, porosity is achieved by the formation of ice crystals according to a rather slow crystallisation process leading to the collapse of CNF and XG to form the walls of aerogels that are thus a dense material. Thus, our procedure results in very low porosity and it is likely that the specific surface of aerogels are very low.  For instance, we already evaluated the specific surface area of CNF aerogels obtained by freeze-drying (Jiménez-Saelices et al, 2017, J. Sol-Gel Sci. Techno, 84 (3), 475-485; Jiménez-Saelices et al, 2017, Carbohydrate Polymers, 157, 105-113) and in all the case the specific surface areas were found to be in the range of 1m2/g which is a very low value. Our purpose was not to optimize the surface area which would have need to applied specific drying procedure such super-critical drying or spray-freeze drying; but to evaluate the water absorption capacities and the shape recovery which can be targeted with directional freeze-drying.

  • English of the manuscript need to be revised carefully.

The manuscript has been carefully edited.

Reviewer 3 Report

First of all, there is big mess with abbreviations CNF and CNC are explined but what are NCF and NCC. 

Are the used cellulose fibrils TEMPO modified - is is not clearly mentioned? 

Also the description of freezing is not well explained - hard to understand how this directional freezing was achieved.

Authors contribution is also little messi.

Author Response

Reviewer 3

 Comments and Suggestions for Authors

First of all, there is big mess with abbreviations CNF and CNC are explained but what are NCF and NCC. 

There was indeed some mistakes with the abbreviations for cellulose nanofibrils and cellulose nanocrystals. CNF and CNC abbreviations for cellulose nanofibrils (CNF) and cellulose nanocrystals, respectively (CNC), were now used all over the text.

Are the used cellulose fibrils TEMPO modified - is is not clearly mentioned? 

In this work, we used TEMPO cellulose nanofibrils by Maine University and aqueous dispersion of CNF were prepared from the as-received nanocellulose with no additional modification. Modification of the “Materials” section (4.1) (Line 325) was done and we hope this preparation was now clear and unambiguous.

Also the description of freezing is not well explained - hard to understand how this directional freezing was achieved.

In order to a better understand of the freezing conditions used in this work, we added a figure in the Supplementary Materials (Line 399) with detailed schemas that explain more clearly the two freezing conditions, i.e. unidirectional or non-directional freezing. This figure is indicated line 150 and 353 of the manuscript.

Authors contribution is also little messi.

The section « Author Contributions » has been rewritten by using the recommended statements of the Gels Journal.

Round 2

Reviewer 2 Report

The authors revised the present manuscript according to my comments. I think the manuscript can be accepted.